# Preferences of South African Adolescents Living with HIV in the Western Cape Province Regarding the Use of Digital Technology for Self-Management

**DOI:** 10.3390/ijerph22070972

**Published:** 2025-06-20

**Authors:** Leonie Weyers, Talitha Crowley, Lwandile Tokwe

**Affiliations:** 1School of Nursing, University of the Western Cape, Cape Town 7535, South Africa; tcrowley@uwc.ac.za; 2HIV Mental Health Research Unit, Division of Neuropsychiatry, Department of Psychiatry and Mental Health, University of Cape Town, Cape Town 7700, South Africa; lwadz.tokwe@uct.ac.za

**Keywords:** Adolescents living with HIV (ALHIV), digital health technology (DHT), self-management, South Africa

## Abstract

Adolescents living with HIV (ALHIV) face significant challenges in self-managing their chronic condition. Digital health technology (DHT) has become increasingly common and understanding ALHIVs’ preferences is essential for developing interventions tailored to this unique population. This study aimed to explore the preferences of ALHIV regarding the use of DHT for self-management. A qualitative research approach with an exploratory and descriptive design was used. Participants were recruited using a purposive sampling method. Data were gathered through six nominal focus groups with 29 participants at two Community Health Centers in the Western Cape Province, South Africa. The participants were ALHIV aged 15–24 years. Discussions focused on current technology usage and the ranking of desired DHT features. The transcripts were analyzed using thematic analysis. Three main themes emerged: (1) everyday usage of digital technology where participants frequently used digital devices for communication, social media, and finding information; (2) the role of digital technology in self-management; a strong interest in digital technology that provides medication reminders, health education, and peer support; and (3) factors influencing digital technology, including the cost of data, limited connectivity, and issues of privacy related to participants’ HIV status. The ALHIV showed a strong willingness to use digital platforms for health information, reminders, and peer support, although concerns about connectivity, data cost, and privacy remain. These findings underscore the need for flexible, user-centered approaches when designing DHT interventions for self-management in South Africa.

## 1. Introduction

Globally, as of 2022, approximately 1.7 million adolescents are living with HIV (ALHIV) [1]. In South Africa, 7.8 million people live with Human Immunodeficiency Virus (HIV), of whom an estimated 360,000 are adolescents aged 10–19 years [2]. Recent data show that HIV prevalence among girls aged 15–24 years is approximately 5.7–8%, whereas the prevalence among their male peers is lower, at 3.1–4% [3,4]. Among adults aged 15–49 years, the 2019 national survey highlights pronounced provincial variation: KwaZulu-Natal (27.0%), Free State (25.5%), Eastern Cape (25.2%), Mpumalanga (22.8%), North West (22.7%), and Gauteng (17.6%) report the highest burdens, while Limpopo (17.2%), Northern Cape (13.9%) and Western Cape (12.6%) record comparatively lower rates [5]. ALHIV in South Africa are part of a vulnerable population who face complex barriers to self-managing their chronic condition [6]. These barriers include adherence to medication, stigma surrounding HIV, and reduced access to healthcare services [6,7]. Failure to address these challenges can lead to decreased treatment outcomes, increased transmission, and higher psychosocial stress among adolescents [8,9].

Digital health technology (DHT), over the last few years, has served as a promising aid to already existing healthcare delivery systems for adolescents [10]. Digital platforms such as mobile health (mHealth) applications, social media, and Short Message Service (SMS) reminders have been shown to improve medication adherence and increase access to reliable health information [11,12]. Self-management involves adolescents actively engaging in their health management tasks, such as medication adherence, symptom monitoring, and accessing health education [13]. Despite this potential, there is limited research focusing on understanding ALHIVs’ specific preferences and perceived barriers to DHT [14]. This is especially true in resource-limited settings where device affordability, internet connectivity, and digital literacy can affect usage [15].

The existing literature suggests that adolescents are receptive to technology-based interventions for health-related information and support, including DHT, such as mobile health applications and social media platforms [8,16]. However, there is a paucity of research that explores the unique experiences of ALHIV in South Africa [17]. Therefore, designing user-centered DHT interventions requires comprehensively exploring adolescents’ everyday use of digital technology. While studies have highlighted the potential of DHT to improve medication adherence and provide peer support [8,9], challenges such as the affordability of mobile data, privacy concerns, and limited internet access remain underexplored within the South African context [18,19].

The barriers to digital health engagement for ALHIV in the Western Cape Province are shaped by a combination of socio-economic factors, such as poverty and limited access to resources, digital access issues like unreliable internet connectivity, and HIV-related stigma, which creates concerns around privacy and the fear of disclosure within digital spaces [8,19].

Because of this gap, we conducted a qualitative, exploratory, descriptive study using constructivist principles. The primary aim of this study was to explore ALHIVs’ preferences regarding the use of DHT for self-management in a resource-limited setting. By identifying the key themes that shape adolescent engagement with digital health technology, the findings may aid future research, healthcare providers, and policymakers in creating interventions that improve adherence and overall well-being among ALHIV.

## 2. Materials and Methods

### 2.1. Study Design

A qualitative research approach was employed. An exploratory and descriptive study design was adopted to capture the preferences of ALHIV regarding the use of DHT. A constructivist paradigm guided the research, focusing on participants’ subjective experiences.

### 2.2. Setting

The study was conducted in two Community Health Centers (CHCs) in the Cape Metropole of the Western Cape Province, South Africa. The CHCs were selected due to their location in low-income, highly populated areas of the Western Cape with significant HIV prevalence. The number of ALHIV, aged 10 to 24 years, attending Crossroads CHC was 500, while the number at Mitchells Plain CHC was 117 in total. The CHCs serve vulnerable populations, including ALHIV who face barriers such as limited resources, HIV related sigma, and healthcare access challenges, making the CHCs the ideal setting for this study.

### 2.3. Participants

The World Health Organization defines adolescence as the period between 10 and 19 years of age; however, Sawyer et al. (2018) propose an extended definition that includes individuals up to 24 years of age [20,21]. A purposive sampling method was used to select ALHIV aged 15–24 years, all of whom are receiving antiretroviral treatment at either of the two clinics. This age range was selected as it represents a period of increasing responsibility for health management. Recruitment took place on Mondays from 08:00 to 09:00 at Crossroads CHC and on Fridays during the same time at Mitchell’s Plain CHC. The recruitment period lasted from 19 June 2023 to 8 March 2024.

Six focus groups were conducted, whereby four were conducted with participants attending the first CHC, and two were conducted with participants attending the second CHC (N = 29). Table 1 below highlights the participants’ demographic characteristics. Participants were divided into groups based on their availability on the scheduled recruitment days at each clinic, as these were the groups they were comfortable with. In qualitative focus group studies, data saturation is usually reached with between four and eight focus groups, especially if the participants are from a homogeneous group [22]. This sample allowed for rich, meaningful insights contributing to information power and thematic saturation.

### 2.4. Data Collection

Nominal focus groups are structured discussions where participants first individually brainstorm ideas, followed by the group sharing and ranking of ideas that arose during the group discussion [23]. Nominal focus groups ensure equal participation while generating diverse perspectives [23,24]. The nominal focus groups consisted of a two-phase approach. An open discussion of current technology was facilitated, followed by the nominal group ranking of desired DHT features. Each session commenced with an open-ended question to create a comfortable atmosphere conducive to open communication. The initial questions explored participants’ general technology use, their preferences, and their daily interactions with digital devices, setting the stage for more in-depth discussions. For the nominal group technique, participants first generated ideas independently, listing features they found important on a piece of paper based on their earlier discussions. All similar ideas were grouped on a whiteboard. These ideas were then shared with the group and clustered into themes, with the research team guiding the process to ensure clarity and avoid duplication. Subsequently, participants ranked these themes according to their perceived importance through a structured voting process, aiming to build consensus on the most valued features.

The focus group sessions were facilitated by the first author and one or both of the co-authors. The sessions lasted on average 120 minutes and were offered in English, Afrikaans, and *isiXhosa*; translators were available to assist.

To ensure qualitative rigor, Lincoln and Guba’s (1985) trustworthiness criteria were applied [25]. Credibility was enhanced through prolonged engagement, participant checking, and debriefing with the research team. Dependability was ensured by maintaining an audit trail and systematic coding with ATLAS.ti software (version 2024). Confirmability involved reflexive journaling and data triangulation. Transferability was supported by detailed descriptions of the study context, enabling applicability to similar low-resource settings. Data collection took place between 9 September 2023 and 16 March 2024. The first focus group was held on 9 September 2023, the second focus group on 14 October 2023, and the third focus group on 21 October 2023. The fourth focus group was held on 18 November 2023, the fifth focus group on 2 December 2023 and the last focus group on 16 March 2024.

### 2.5. Data Analysis

The audio recordings were transcribed verbatim by the author, and the data were analysed using the ATLAS.ti (version 2024) software, which assisted with initial coding and identifying themes. The transcripts and audio files were shared with the research team for quality assurance. This assisted in minimizing researcher bias. The initial codes were generated through an iterative process. The research team held meetings to discuss the initial codes, reconcile differences, and refine the codes into a consistent coding framework. This study followed the reflexive approach as it best aligns with qualitative studies that explore complex, subjective experiences [26,27]. Using Braun and Clark’s (2021) six phases of reflexive thematic analysis, the researchers first familiarized themselves with the data [26,28]. Thereafter, initial codes were generated as explained above, followed by theme identification and refinement. The themes were reviewed by the research team, defined, and named. Finally, a detailed write-up was created, incorporating direct quotations to illustrate the findings.

Data from the nominal ranking of features were collated in Microsoft Excel and analyzed descriptively. The participants’ votes for each feature were ranked through an anonymous voting process, and the frequency of votes was calculated. The results were then tabulated, with scores reflecting the percentage of participants who selected each feature. Themes were analyzed and presented separately as adolescents’ preferred features for DHT.

### 2.6. Ethical Considerations

This study adhered to rigorous ethical principles to ensure the protection, dignity and well-being of all participants. Ethical clearance was obtained from the University of the Western Cape Biomedical Research Ethics Committee (Ref: BM23/6/14), and permission was granted by the Western Cape Department of Health, Ref: WC202308038. Informed consent was obtained from all participants, with additional parental or guardian consent required for participants under the age of 18 years old. The informed consent process included a clear explanation of the study’s purpose, procedures, potential risks, and benefits to ensure that participants could make an informed decision about their involvement. Participation was completely voluntary, and participants were informed of their right to withdraw at any stage without consequences or impacts on their healthcare services.

Confidentiality and anonymity were ensured by removing personal identifiers and securely storing de-identified data. This was ensured by assigning participants numbers at the beginning of the focus groups. The research team used the numbers assigned to request clarity from the participants. Compensation of ZAR 50 was provided to cover transport to and from the research site. Participants signed their names on a list after receiving the compensation. The focus groups were conducted in a safe environment with psychological support from the research team available. Discussions were held in English and Xhosa with translators. Data protection followed the University of the Western Cape’s policies, and participant selection ensured diverse representation. Ethical safeguards upheld research integrity and highlighted participant well-being.

## 3. Results

Through reflexive thematic analysis of the nominal focus group data, three main themes emerged, each reflecting how ALHIV in South Africa engage with digital technology for health management (Table 2).

### 3.1. Theme 1: Everyday Usage of Digital Technology

Participants described digital technology as an integral part of their daily lives, such as using it for communication, education, entertainment, and health information. Smartphones were the most commonly used devices, with some participants relying on shared devices due to financial constraints.

#### 3.1.1. Subtheme 1.1: Integrating Digital Technology into Daily Life

Most participants owned or had access to smartphones, which they used for various purposes, including staying connected with family and friends, researching schoolwork, and entertainment. However, some participants relied on shared devices.


*“I use my smartphone for everything, from chatting with friends to doing my homework.”*
(Participant 1, aged 20, Focus group 1)


*“Sometimes I’m using my sister’s laptop.”*
(Participant 3, aged 19, Focus group 2)

WhatsApp and Facebook were the most frequently used platforms for staying in touch with family and peers.


*“Because I don’t even know how to use the other apps. That’s why I like Facebook and WhatsApp. That’s all I like.”*
(Participant 3, aged 19, Focus group 2)

Google was frequently used for schoolwork, and some participants mentioned using laptops or computers at school for specific subjects.


*“I once searched for an essay for my history, so I took notes from there, and then I did my essay because Facebook has a free mode.”*
(Participant 1, aged 16, Focus group 3)


*“We are using computers, for example, the subject I do at school I’m doing computer application technology, and we also do the practical that can teach us the technology.”*
(Participant 3, aged 19, Focus group 2)

Participants frequently engaged with TikTok, YouTube, and gaming applications for entertainment.


*“I love TikTok and YouTube, even Facebook.”*
(Participant 2, aged 15, Focus group 6)


*“Candy Crush and Bubble Shooter and 2248 are about numbers”*
(Participant 4, aged 21, Focus group 2)

#### 3.1.2. Subtheme 1.2: Popular Digital Platforms and Their Uses

As mentioned before, WhatsApp and Facebook were the most commonly used platforms, primarily due to their accessibility and affordability.


*“It can be helpful by creating a WhatsApp group and also on Facebook creating a group because I don’t have data so there’s a free mode on Facebook.”*
(Participant 4, aged 21, Focus group 2)

Many participants preferred YouTube and TikTok for entertainment and learning purposes.


*“And I also use Google or YouTube when I’m watching movies.”*
(Participant 3, aged 19, Focus group 2)


*“Most of the time I’m using TikTok because I enjoy watching videos and also WhatsApp, Facebook and YouTube.”*
(Participant 1, aged 19, Focus group 2)

### 3.2. Theme 2: Role of Digital Technology in Self-Management

Digital tools were used to support personal health management, particularly through reminders and information seeking.

#### Subtheme 2.1: Using Technology for Personal Health Management

Participants highlighted the role of digital tools, particularly smartphones, in setting reminders for medication adherence and other health-related activities.


*“I use the phone to set alarms for my meals because I eat four times a day and to take my pills. The phone helps me to get a reminder to take my medication.”*
(Participant 2, aged 20, Focus group 4)


*“I use my phone only for waking up as an alarm and also as a reminder for taking medication.”*
(Participant 5, aged 20, Focus group 4)

Some participants used their phones to search for health-related information, such as diet recommendations and minor health concerns.


*“I use the phone to search for healthy food.”*
(Participant 2, aged 20, Focus group 4)


*“I had a boil on my armpit, so I went to Google and searched how to heal it and what caused it.”*
(Participant 1, aged 16, Focus group 3)

One participant noted using an Artificial intelligence (AI) chatbot on WhatsApp to discuss health concerns privately.


*“There is an AI on my WhatsApp. The AI gives me advice, and I can talk to it about stuff that I can’t talk to my mother about.”*
(Participant 2, aged 18, Focus group 3)

### 3.3. Theme 3: Factors Influencing Digital Technology Use

Participants identified several barriers and facilitators of digital technology use, including connectivity, privacy concerns, and the role of family and peers.

#### 3.3.1. Subtheme 3.1: Connectivity and Internet Access

Limited mobile data and unreliable Wi-Fi were significant challenges, especially during loadshedding.


*“When there is load shedding, there’s a problem with the network, and also some people have Wi-Fi, so when there’s load shedding, they can’t use their Wi-Fi because the Wi-Fi is off, so there’s no Internet.”*
(Participant 4, aged 21, Focus group 2)


*“I’m using Wi-Fi, but when I’m not at home, I’m using my phone data.”*
(Participant 6, aged 20, Focus group 4)

#### 3.3.2. Subtheme 3.2: Privacy and Digital Security

Privacy concerns influenced how participants used digital tools. Some preferred security features like passwords and biometric authentication.


*“If I give you my phone to do the research and then there’s going to be like, WhatsApp and all of those apps, then I put like a fingerprint to get into it because you need my fingerprint to get into it. That way, I will feel safer.”*
(Participant 4, aged 15, Focus group 4)

#### 3.3.3. Subtheme 3.3: Assessing Information Credibility

Participants expressed mixed opinions about the reliability of online health information. While some trusted digital sources, like Google, for schoolwork and health-related searches, others questioned the accuracy of the information provided.


*“I’m not actually sure if I must trust the Internet because when I must do my research tasks, they give me answers to other questions that I did not ask, but then if I must write that down and go back to school, they say no, this is not right, do this over.”*
(Participant 4, aged 15, Focus group 4)

Some participants preferred verifying information through direct consultation with knowledgeable individuals rather than relying on online sources.


*“I don’t like this, so I would rather go ask the person who does the job himself how and why.”*
(Participant 4, aged 15, Focus group 4)

Despite skepticism from some, others found online information beneficial for verifying facts and completing schoolwork.


*“I will say that I trust it because when someone tells me about something, and then I am not sure about it, I’ll go to Google and Google it and see if it is true or not. So, I trust it.”*
(Participant 4, aged 21, Focus group 2)


*“The schoolwork I was also getting right because of Google.”*
(Participant 2, aged 21, Focus group 5)

Health-related searches often provide incomplete or unclear information, leading to uncertainty about its reliability.


*“Because everything that I’m searching for, it is giving a clue but not the real thing.”*
(Participant 1, aged 20, Focus group 1)

#### 3.3.4. Subtheme 3.4: Family and Peer Dynamics

Family played a supportive role in ensuring medication adherence and promoting digital engagement.


*“My family is very supportive. All of them would like phone just to remind me and say, listen, you must not forget about you tablets”*
(Participant 4, aged 15, Focus group 4)

Participants also highlighted how peer influence shaped their engagement with certain platforms.


*“Yes, especially with TikTok. I didn’t like this before and then my dad said I must download it because there’s a child who likes to cook like you, so then I downloaded it.”*
(Participant 2, aged 21, Focus group 5)


*“If it’s coming from friends, I ignore it, but if it’s coming from one of the family members, that’s when I will start to give more attention to it.”*
(Participant 5, aged 19, Focus group 1)

The themes extracted from the nominal ranking of participants’ preferences for DHT are presented in Table 3 and Figure 1.

## 4. Discussion

This study explored the preferences of ALHIV regarding the use of DHT for self-management and identified three major themes, as well as a list of preferred features.

Everyday usage of digital technology: The use of digital technologies among ALHIV for communication, education, and entertainment reflects their significant integration into daily life. This study has similar findings to L’Engle et al. [29], who observed that digital platforms are crucial for accessing health-related information, especially for adolescents dealing with stigma or fear of disclosing their status. Additionally, Chory et al. [30] noted the role of platforms such as WhatsApp in providing emotional support and advice, which is vital for coping with their condition. Bhana et al. [31] further underscore the importance of digital technology in facilitating access to mental health resources, which is particularly relevant for adolescents facing chronic health conditions like HIV.

The role of digital technology for self-management: Digital technology serves a critical role in self-management for ALHIV, particularly through functions like medication reminders and health monitoring. This aligns with global trends where mHealth interventions—such as the M-TIBA platform, which stands for “Mobile Care” in Swahili, in Kenya—enhance healthcare engagement by simplifying access to health services and managing healthcare expenses through mobile devices [11,32]. Furthermore, systematic reviews indicate that mobile health applications and SMS reminders significantly improve antiretroviral therapy (ART) adherence and overall engagement in care [33,34]. However, digital literacy remains a challenge for many adolescents, requiring additional support to maximize the benefits of these interventions [35].

Factors influencing digital technology use: Despite the benefits of digital technology, several barriers, such as affordability and accessibility, impact its effective use among ALHIV. Many adolescents rely on shared devices, which can limit consistent access and compromise privacy. This was also highlighted by Gunnlaugsson et al. [36]. Additionally, Allen et al. [37] discuss how socio-economic inequalities can restrict access to digital tools, stressing the need for equitable digital health solutions. Abiodun et al. [34] observed that while SMS reminders can enhance ART adherence, concerns over privacy and data security remain significant, particularly in resource-limited settings.

Participant preferences for digital health technology: Participants expressed a strong preference for digital health technologies that include educational content, reminders, and social interaction. These preferences underscore the need for interactive and multimedia content that engages users effectively while providing health information and social support, as mentioned by Mulawa et al. [38]. The Masakhane Siphucule Impilo Yethu (MASI) study in South Africa demonstrated that a smartphone-based intervention incorporating reminders, motivational messages, and direct access to healthcare professionals was highly valued among ALHIV [16]. Furthermore, Cho et al. [39] suggest that engaging and personalized digital health interventions can significantly enhance user experience and adherence to prescribed health regimens.

This study underscores several possibilities for further development and implementation of DHT. Interventions should adopt user-centered design principles, providing unique reminders and educational components that respect adolescents ‘confidentiality requirements. Additionally, forming partnerships with telecommunication providers can help address data cost challenges by introducing subsidized or zero-rated access, thus reducing connectivity barriers. Finally, including adolescents’ families and local communities in the implementation process while protecting their privacy can enhance both digital literacy and the acceptance of DHT interventions. This study highlights that adolescents are open to using digital tools for health, suggesting that healthcare workers and adolescent programs should incorporate digital technology to support self-management and improve adolescent engagement in care.

A strength of this study is the use of nominal focus groups, which gave a collective ranking of preferred DHT features and a rich perspective on adolescents’ day-to-day technology usage. However, because the findings are gathered from only two CHCs located in one province, transferability to other contexts may be constrained. Therefore, future research should consider broadening the sample to include more demographically and geographically diverse populations. It is important to note that the Western Cape and Northern Cape Provinces report lower HIV prevalence compared to other provinces in South Africa [5]. This variation may be attributed to factors such as stronger health system infrastructure, lower population density, and more consistent ART coverage [40]. These contextual differences highlight the importance of tailoring DHT interventions to specific regional characteristics. An additional limitation is that many participants relied on shared digital devices due to limited financial resources, which may have affected their ability both to consistently access and use digital health tools and to express their preferences regarding these tools. Future studies could strengthen the evidence base by examining how access to personal digital devices affects engagement with DHT and exploring potential differences across narrower age bands. Moreover, integrating treatment characteristics and clinical indicators, such as duration of ART, ART adherence, and viral load suppression, would provide a more comprehensive understanding of the potential impact of digital interventions on treatment outcomes among ALHIV.

## 5. Conclusions

ALHIV in the Western Cape, South Africa, value DHT that facilitates treatment reminders, secures information, and has peer support. However, addressing privacy, costs, and limited connectivity is vital for its successful implementation. Compared to adolescents in other countries, South African participants placed greater emphasis on affordability, privacy, and the use of shared digital devices. These results offer practical guidance for designing DHT interventions that resonate with adolescents’ lived realities.

## Figures and Tables

**Figure 1 ijerph-22-00972-f001:**
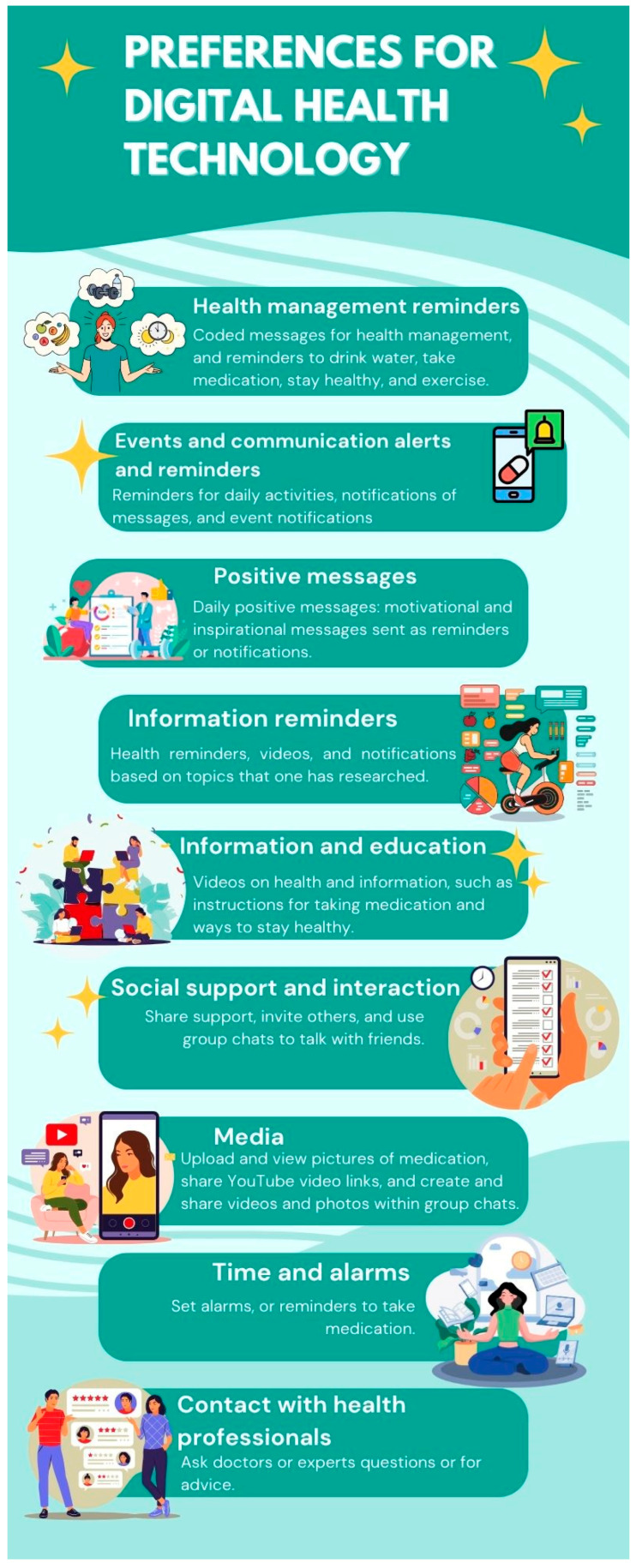
Infographic of adolescents’ preferences for DHT.

**Table 1 ijerph-22-00972-t001:** Participants’ demographic characteristics.

Category	Sub-Category	Total (N = 29)
Age (in years)	15–19	17 (58.6%)
	20–24	12 (41.4%)
Sex	Female	19 (65.5%)
	Male	10 (34.5%)
Highest level of education	Middle School Grades 8–10	10 (34.5%)
	High School Grades 11–12	15 (51.7%)
	Post-School Education	4 (13.8%)

**Table 2 ijerph-22-00972-t002:** Themes, subthemes, and codes.

Theme	Subtheme	Code
1. Everyday usage of digital technology	1.1 Integrating digital technology into daily life	Smartphone usage
		Shared devices
		Television viewing
		Computer usage
		Communication through technology
		Using technology for education
		Technology for entertainment purposes
		Accessing health information through technology
	1.2 Popular digital platforms and their uses	Using Facebook
		Playing games
		Using Google
		Using Instagram
		Using TikTok
		Using WhatsApp
		Using YouTube
2. The role of digital technology in self-management	2.1 Using technology for personal health management	Using a smartphone to set alarms for health reminders
		Using a smartphone to track health metrics
3. Factors that influence digital technology use	3.1 Connectivity and internet access	Power outages affecting Wi-Fi connectivity
		Availability of mobile data
		Wi-Fi availability
		Signal accessibility
	3.2 Privacy and digital security	Feeling secure with applications
		Feeling secure with the internet
	3.3 Assessing information credibility	Trust in online information
	3.4 Family and peer dynamics	Emotional support from family members
		Family attendance at clinic appointments
		Peer influence on technology use
		Social media peer pressure
		Clinic and medication reminders from family
		Family restrictions on technology
	3.5 Digital technology literacy	Lack of knowledge of specific apps
		Parental lack of technology literacy

**Table 3 ijerph-22-00972-t003:** Themes and participant preferences for DHT.

Theme	Participant Ideas	Score (N = 29)
Information and education	Videos on health and information, such as instructions for taking medication and ways to stay healthy.	13 (44.8%)
Health management reminders	Coded messages for health management, and reminders to drink water, take medication, stay healthy, and exercise.	12 (41.4%)
Social support and interaction	Share support, invite others, and use a group chats to talk with friends.	6 (20.7%)
Events and communication, alerts, and reminders	Reminders for daily activities, notifications of messages, and notifications that an event is about to start.	5 (17.2%)
Positive messages	Daily positive messages: motivational and inspirational messages sent as reminders or notifications.	4 (13.8%)
Media	Upload and view pictures of medication, share YouTube video links, and create and share videos and photos within group chats.	4 (13.8%)
Information reminders	Reminders about what one should do or videos on health, and notifications on topics that one has researched to obtain more information.	3 (10.3%)
Time and alarms	Set alarms or reminders to take medication.	3 (10.3%)
Contact with health professionals	Ask doctors or experts questions or for advice.	2 (6.9%)

## Data Availability

De-identified transcripts and nominal ranking data are available from the corresponding author upon reasonable request, subject to ethical and confidentiality constraints.

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
