# Peer review of "Preferences of South African Adolescents Living with HIV in the Western Cape Province Regarding the Use of Digital Technology for Self-Management"

_ijerph, 2025, doi:10.3390/ijerph22070972_

Round 1
Reviewer 1 Report
Comments and Suggestions for Authors
The authors conducted a qualitative descriptive study that exploring the preferences of adolescents living with HIV regarding the use of digital health technology for self-management in a limited resource setting. The study identified three major themes as well as a list of preferred features. The authors provide a well-written paper with thoughtful study design and discussion. Regarding to the research strategy and findings, reviewers only have minor comments that should be addressed prior to publication.
- Table 1 - Add unit (years) after "Age"
- Limitation - One more limitation is that study acknowledges that participants often rely on shared devices due to financial constraints, which could impact their engagement with digital health technology.
Reviewer 2 Report
Comments and Suggestions for Authors
Please see the attachment.

The text requires stylistic processing, because sometimes it is difficult to understand what the authors mean.
Reviewer 3 Report
Comments and Suggestions for Authors
- Consider clarifying whether “Participant Number” in the demographics table refers to the count of participants or is intended as a de-identifying code. Does this refer to the number of participants?
- Consider aggregating demographics into categories (gender, age groups, education, focus group) and reporting each as n (%) in a single table rather than listing individual attributes.
- Consider replacing “Participant X, aged 20, Focus Group 1” (pages 6-8) line with a neutral pseudocode to minimize identifiability.
Round 2
Reviewer 2 Report
Comments and Suggestions for Authors
Please see the Attachment.

Reviewer 3 Report
Comments and Suggestions for Authors
Thank you for making the updates.
